# TIMP1 and TIMP2 Downregulate TGFβ Induced Decidual-like Phenotype in Natural Killer Cells

**DOI:** 10.3390/cancers13194955

**Published:** 2021-10-01

**Authors:** Adriana Albini, Matteo Gallazzi, Maria Teresa Palano, Valentina Carlini, Riccardo Ricotta, Antonino Bruno, William G. Stetler-Stevenson, Douglas M. Noonan

**Affiliations:** 1Laboratory of Vascular Biology and Angiogenesis, IRCCS MultiMedica, 20138 Milan, Italy; 2Immunology and General Pathology Laboratory, Department of Biotechnology and Life Sciences, University of Insubria, 21100 Varese, Italy; m.gallazzi7@uninsubria.it (M.G.); douglas.noonan@uninsubria.it (D.M.N.); 3Laboratory of Innate Immunity, Unit of Molecular Pathology, Biochemistry and Immunology, IRCCS MultiMedica, 20138 Milan, Italy; mariateresa.palano@multimedica.it; 4Unit of Molecular Pathology, Biochemistry and Immunology, IRCCS MultiMedica, 20138 Milan, Italy; valentina.carlini@multimedica.it; 5IRCCS MultiMedica, 20099 Sesto San Giovanni, MI, Italy; riccardo.ricotta@multimedica.it; 6Extracellular Matrix Pathology Section, Laboratory of Pathology, National Cancer Institute, National Institute of Health, Bethesda, MD 20892, USA; sstevenw@mail.nih.gov

**Keywords:** TIMP-1, TIMP-2, TGFβ, Natural Killer cells, decidual-NK cells, tumor microenvironment, cancer, innate immunity, inflammation, angiogenesis

## Abstract

**Simple Summary:**

Cancer patients are characterized by NK cells with altered surface markers, such as CD56 brightness, CD9, CD49a (pro-angiogenic) and PD-1, and TIM-3 (exhaustion), that favor immune escape. Transforming growth factor-beta (TGFβ) is a major tumor-derived cytokine that favors cancer growth and supports pro-angiogenic activities in NK cells by inducing pro-angiogenic molecules. TIMP-1 and TIMP-2 play a crucial role in extracellular matrix (ECM) regulation, wound healing, pregnancy and cancer, and there is increasing evidence that they are immune-modulatory. We found that recombinant TIMP-1 and -2 can partially contrast the induction of pro-tumor/pro-angiogenic decidual-like polarization of NK cells by TGFβ.

**Abstract:**

Natural Killer (NK) cells have been found to be anergic, exhausted and pro-angiogenic in cancers. NK cell from healthy donors, exposed to TGFβ, acquire the CD56^bright^CD9^+^CD49a^+^ decidual-like-phenotype, together with decreased levels of NKG2D activation marker, increased levels of TIM-3 exhaustion marker, similar to cancer-associated NK cells. Tissue inhibitors of metalloproteases (TIMPs) exert dual roles in cancer. The role of TIMPs in modulating immune cells is a very novel concept, and the present is the first report studying their ability to contrast TGFβ action on NK cells. Here, we investigated the effects of TIMP1 and TIMP2 recombinant proteins in hindering decidual-like markers in NK cells, generated by polarizing cytolytic NK cells with TGFβ. The effects of TIMP1 or TIMP2 on NK cell surface antigens were determined by multicolor flow cytometry. We found that TIMP1 and TIMP2 were effective in interfering with TGFβ induced NK cell polarization towards a decidual-like-phenotype. TIMP1 and TIMP2 counteracted the effect of TGFβ in increasing the percentage of CD56^bright,^ CD16^−^, CD9^+^ and CD49a^+^, and restoring normal levels for TIMP 1 and 2 also inhibited decrease levels of the activation marker NKG2D induced by TGFβ and decreased the TGFβ upregulated exhaustion marker TIM-3. NK cell degranulation capabilities against K562 cells were also decreased by TGFβ and not by TIMP1 or TIMP2. TIMP1 treatment could partially restore degranulation marker CD107a expression. Treatment with recombinant TIMP-1 or TIMP-2 showed a trend, although not statistically significant, to decrease CD49a^+^ and TIM-3+ expression and increase NKG2D in peripheral blood NK cells exposed to conditioned media from colon cancer cell lines. Our results suggest a potential role of TIMPs in controlling the tumor-associated cytokine TGFβ-induced NK cell polarization. Given the heterogeneity of released factors within the TME, it is clear that TGFβ stimulation represents a model to prove TIMP’s new properties, but it cannot be envisaged as a soloist NK cell polarizing agent. Therefore, further studies from the scientific community will help defining TIMPs immunomodulatory activities of NK cells in cancer, and their possible future diagnostic–therapeutic roles.

## 1. Introduction

Tissue inhibitors of metalloproteases (TIMPs) were discovered in the 1980s [1], mostly for their function, as the name says, in inhibiting the enzymatic activity of metalloproteases (MMPs). There are four known members, TIMPs 1–4, with both similarities and many differences in functions, depending also on the microenvironment stimuli [2]. TIMPs are proteins of 184–194 amino acids and ~21 kDa in molecular weight, showing ~40% identity (~60% similarity) in sequence. TIMP2 and TIMP4 are structurally related, sharing most similarities with one another, while TIMP1 represents the most unique member of the TIMP protein family. We were the first to discover the role of TIMP2 in inhibiting tumor invasion [3].

TIMPs have been found to exert biological activities independent of MMP inhibition; TIMP perturbations lead to complex and unexpected context- and tissue-specific biological outcomes [2]. Roles of TIMPs, besides inhibition of invasion and metastases, have been identified in anti-angiogenesis, modulation of stem cell properties, neuronal regulation, and monocyte functions [2,4].

In contrast to being a MMP inhibitor and therefore an anti-invasive agent, clinical studies have shown a paradoxical association of high TIMP1 expression with a poor prognosis and more advanced stage tumors in a variety of cancer patients (i.e., lung, brain, prostate, breast, colon, and endometrial) [2,5,6]. Lack of TIMP1 (but not TIMP2) immunostaining is associated with a favorable prognosis in patients with node-positive high-grade breast carcinoma [2,7]. TIMP2 is an anti-angiogenic protein [4,8], and tumor-bearing timp2−/− mice show significantly increased inflammatory cells, myeloid-derived suppressor cell (MDSC; CD11b^+^ and Gr-1^+^) and endothelial cells (ECs) in tumors compared to the wild type [9]. TIMP2, but not TIMP1, secreted by monocyte-like cells, is a potent suppressor of invadopodium formation in breast [10] and pancreatic [11] cancers. This shows that TIMPs can behave as stress-response genes similar to “alarmins” [12]. Alarmins are danger-signal molecules produced by the cell under stress (i.e., hypoxia, nutrient starvation, inflammation), and they are endogenous, immune-activating proteins/peptides released as a result of degranulation or in response to immune induction [12]. Non-MMP-inhibitory and oncogenic functions of TIMP1 are mediated by the initiation of intracellular signaling via its tetraspanin cell surface receptor CD63 [13].

The TGFβ signaling pathway governs key cellular processes under physiologic conditions and is deregulated in many pathologies, including cancer [14,15,16]. TGFβ is a multifunctional cytokine that acts in a cell- and context-dependent manner as a tumor promoter or tumor suppressor [14,15,16]. TGFβ-induces TIMP1 from hepatic stellate cells, subsequently bound to CD63, leading to FAK activation in hepatocellular carcinoma cells, suggesting that TIMP1 can function as a mediator of TGFβ-regulated crosstalk between stromal and cancer compartments [2,17].

On the other hand, TIMP2 functions almost exclusively in an anti-tumorigenic manner. It has been shown that TIMP2 inhibits in vivo VEGF-induced angiogenic responses and primary tumor growth of human lung xenografts, as well as inhibiting metastasis in an orthotopic, murine triple-negative breast cancer (TNBC) model [18,19,20].

ILCs (innate lymphoid cells) represent a recently identified heterogeneous family of mononuclear hematopoietic cells found mostly in solid tissues [21,22]. Based on their lymphoid morphology, surface antigens, transcription factor expression and cytokine production (Th1, Th2 and Th17-like), ILCs have been classified into three major groups, termed ILC1, ILC2 and ILC3 [22,23]. Whether ILCs can be defined as friends or foes in cancer insurgence and progression is still a matter of debate [21,22,24].

NK cells are innate lymphoid cells involved in tumor immunosurveillance by inducing cancer cell lysis both directly (via perforin/granzyme system) or indirectly (via secretion of TNFα and IFNγ) [21,25]. NK cells develop altered phenotypes in many cancer types, which includes acquisition of anergy and cell exhaustion (increased levels of PD-1 and TIM-3) [26,27,28]. Apart from anergy, we found that tumor-associated peripheral blood NK cells (TANKs) in non-small cell lung cancer (NSCLC), pleural effusions and colorectal cancers (CRC) and prostate cancer patients acquire pro-angiogenic phenotype and functions [29,30,31,32,33]. We observed that tumor-associated (TANKs) and tumor-infiltrating (TINKs) NK cells in cancer patients can acquire phenotype and functions similar to NK cells in the developing decidua [34,35,36,37]. The decidual NK (dNK) cells are characterized by a CD56^superbright^CD16^neg^ phenotype and by the presence of CD9 (a member of the tetraspanin protein family that plays a role in cell adhesion and cell motility) and CD49a (an integrin alpha 1 subunit that binds collagen and laminin) [35,38]. These dNK cells are poorly cytotoxic and are necessary to remodel spiral arteries during pregnancy [34,35,39,40]. Within the decidua, Natural Killer (NK) cells play a critical role in the tolerance of the developing fetus, the modulation of angiogenesis and decidualization, all necessary processes associated with pregnancy. dNKs produce angiogenic factors: VEGF, PlGF and IL-8 (CXCL8) [35,39,40,41], angiogenin [40,41] and angiopoietin 1 and 2 [35,41]. They also support tissue remodeling through the expression of matrix metalloproteinases (MMP-9). MMPs and TIMPs expression was observed in both decidual tissues and in dNK cells [42]. It was found that a wide variety of MMPs are expressed at the human fetal–maternal interface, supporting the highly invasive cellular activity and ECM remodeling that occurs during placental development. Interestingly, some NKG2D ligands (MICA, MICB, ULBP-2 and ULBP-3) are secreted from NSCLC [43], osteosarcomas [44] and gastric [45] cancer cells and may be shed by MMP-mediated cleavage. For example, upregulation of MMP activity can induce a downregulation of the expression of NKG2D ligands in gastric cancer cells, leading to lower-level cancer susceptibility to NK cells [45]. Therefore, TIMPs could be relevant in NK cell recognition activities. Supernatants derived from decidual NK cells show pro-angiogenic action in vitro and in vivo and can significantly increase tumor growth and angiogenesis [39]. We have found that CRC TANKs expressed CD9 and CD49a, as well as increased VEGF and CXCL8, production and release when compared with healthy controls NK cells, behaving similarly to dNK [30]. We have also described a paradoxical increased expression of TIMP1 and TIMP2 in CRC NK. All these features support the rationale that pro-angiogenic tumor-associated NK cells, as “onco-fetal” NK, resume features of embryo “nurturing” cells [37].

Transforming growth factor-β (TGFβ) is associated with dNK cell polarization [46,47] and is present in the tumor microenvironment (TME). A combination of TGFβ, hypoxia and a demethylating agent induces a dNK-like phenotype in healthy donor NK cells [48]. TGFβ converts NK cells into the intermediate ILC-1 cell population, which is unable to control local tumor growth and metastasis [49]. We demonstrated that TGFβ can also generate decidual-like NK cells, able to produce VEGF and PlGF [32].

Here, we show the ability of TIMP1 and TIMP2 to contrast the TGFβ-induced decidual-like polarization of cytolytic NK cells. In particular, TIMP1 and TIMP2 counteracted the effect of TGFβ in increasing the percentage of CD56^bright^, CD16^−^, CD9^+^ and CD49a^+^, restoring normal levels. TIMPs 1 and 2 also inhibited decreased levels of the lytic marker NKG2D induced by TGFβ and hindered the TGFβ upregulated TIM-3.

NK cell degranulation abilities on K562 cells were also decreased by TGFβ but not by TIMP1 or TIMP2. However, only TIMP1 could also partially restore degranulation marker CD107a, while TIMP2 showed a small but not significant trend.

In preliminary experiments, we treated NK cells with conditioned media (CM) of colon cancer cell lines. CD9, CD49a and TIM3 expression was increased by CM, as previously reported, while TIMP1 and 2 showed a trend in counteracting the TGFβ effects. Compared to TGFβ, the cytokine IL-6 was not able to induce a decidual-like polarization in cytolytic NK cells.

Our results suggest a potential role of TIMPs in controlling the angiogenic switch in tumor-associated NK cells induced by TGFβ and provide the rational for the possible use of TIMPs in the re-education of anergic/pro-angiogenic NK cells in cancer patients.

## 2. Materials and Methods

### 2.1. Preparation of the Recombinant TIMP-1 and TIMP-2

Recombinant TIMP1 and TIMP2 with 6X His-epitope tag were expressed, purified and tested negative for Endotoxin in the MTBM assay (Animal Health Diagnostic Laboratory, NCI-Frederick, Frederick, MD, USA). TIMP1 and TIMP2 preparations showed >98 purity. Protein concentration was determined by A280, bicinchoninic acid (BCA) assay and sandwich ELISA assays. Lyophilized TIMP1 and TIMP2 powder was reconstituted in Hanks Balanced Salt Solution (HBSS), sterile-filtered (0.22 μm) and stored at −80 °C until use [50]. Preliminary experiments were performed to choose the best concentration of the recombinant TIMPs for NK cell treatment.

### 2.2. Isolation of Mononuclear Cells from Whole Blood of Healthy Donors

Twelve milliliters of heparinized whole blood, collected from healthy donors, was diluted with PBS 1:1 (v/v), then subjected to a density gradient stratification with Ficoll Histopaque-1077 (Sigma Aldrich, St. Louis, MO, USA) at 500× *g* for 20 min. The white ring interface, composed of total mononuclear cells (MNCs), was collected, washed twice in PBS, then used for subsequent experiments for NK cell polarization and treatments. Human samples were collected from healthy donors, enrolled within studies approved by the institutional review board ethics committees (protocol no. 0024138 04/07/2011, University of Insubria, Italy and protocol no. 10 2 10/2011, IRCCS MultiMedica, Milan, Italy) and according to the Helsinki Declaration of 1975 as revised in 2013. All subjects included in the study signed the informed consent, in accordance with the Helsinki Declaration of 1975 as revised in 2013.

### 2.3. NK Cell Polarization and Treatments with TIMPs

A total of 3 × 10^6^ mononuclear cells (MNCs), from peripheral blood of healthy donors, were polarized with TGFβ (10 ng/mL) alone or in combination with TIMP-1 (0.1 μg/mL) or TIMP-2 (0.1 μg/mL) or IL-6 (25 ng/mL) alone. MNCs received treatments at day zero and at 48 h. FACS analysis was performed following 72 h of treatments.

### 2.4. Colon Cancer Cell Line Culture and Maintenance

The colon cancer (CC) HT-29 and the colorectal cancer (CRC) CaCo2 cell lines (ATCC) were maintained in RPMI 1640 medium, supplemented with 10% Fetal Bovine Serum (FBS), (Euroclone, Pero, MI, Italy), 2 mM l-glutamine (Euroclone), 100 U/mL penicillin and 100 μg/mL streptomycin (Euroclone). Conditioned media for NK cell polarization experiments were collected from HT-29 cells. Once cells were 80% confluent, cells were washed for 30 min in serum-free RPMI medium to eliminate serum residuals. Following cell layer wash, cells were maintained in serum-free RPMI medium for 48 h and conditioned media (CMs) were collected. CMs were concentrated using the 3KDa cut-off concentrations (Millipore, Burlington, MA, USA) and quantified by Bradford reagent. Aliquots of 50 μg total protein were prepared and used for NK cell polarization.

### 2.5. Phenotype Characterization of TGFβ or CC/CRC CM Exposed NK Treated with TIMPs

The effects of TIMP1 and TIMP2 on decidual-like NK cells, generated by MNC stimulated by TGFβ, were determined by multicolor flow cytometry, using a BD FACS Fortessa ×20 analyzer, equipped with 5 lasers. A total of 2 × 10^5^ of total MNCs were stained for 30 min at 4 °C with anti-human monoclonal antibodies (mAbs) as follows: PerCP-conjugated anti-CD3 (BW264/56), APC-conjugated anti-CD56 (REA196), FITC-conjugated anti-CD16 (REA589), PE-conjugated anti-CD9 (REA1071), PE-conjugated anti-CD49a (REA1106), PE-conjugated anti-NKG2D (REA1228), PE-conjugated anti-TIM-3 (F38-2E2) and PE-conjugated anti-PD-1 (PD1.3.1.3) (all purchased by Miltenyi Biotec, Bergisch Gladbach, Germany). IL-6 alone was used as control for TGFβ activity. CM (50 µg) from HT-29 or CaCo2 cell lines were also used as stimuli in preliminary experiments.

Following Forward/Side Scatter setting, NK cells were identified as CD3^−^ and CD56^+^ cells (total NK cells). CD16 and NKG2D expression was evaluated on CD3^−^CD56^+^ (total NK) gated cells. Finally, CD56 brightness, the expression of the dNK markers CD9, CD49a, expression of TIM-3 and the expression of the exhaustion markers PD-1 and TIM3 were evaluated on total CD3^−^CD56^+^NK cells.

### 2.6. Degranulation Assay on TGFβ-Polarized NK Cells Exposed to TIMPs

A total of 3 × 10^6^ mononuclear cells (MNCs), from peripheral blood of healthy donors, were polarized in RPMI medium with TGFβ (10 ng/mL), alone or in combination with TIMP-1 (0.1 μg/mL) or TIMP-2 (0.1 μg/mL) or IL-6 (25 ng/mL) alone. MNCs received treatments at day zero and at 48 h. After 72 h, polarized cells were used to detect their ability to degranulate, as detected by the CD107a expression, against the K562 cell line, according to [51,52,53], with minor modifications. Following 72 h of polarization, 2 × 10^5^ MNCs were co-cultured with 2 × 10^5^ K562 (E:T ratio of 1:1) in the presence of anti-CD107a- FITC (BD Biosciences, San Jose, CA, USA, H4A3), Golgi Plug (Brefeldin, BD Biosciences) and Golgi Stop (Monesin, BD Biosciences). MNC alone was used as control to detect basal degranulation activities by NK cells, MNC treated with Ionomycin (500 ng/mL, Sigma Aldrich), PMA (10 ng/mL, Sigma Aldrich), Golgi Plug (Brefeldin, BD Biosciences) and Golgi Stop (Monesin, BD Biosciences), as positive control for non-specific degranulation, while K562 cell alone was used as internal control. Cells were stimulated for 6 h. CD107a expression, as a readout of degranulation activities, was detected by flow cytometry on CD3^+^CD56^+^ total NK cells. Cell degranulation efficiency was finally determined by subtracting the basal degranulation (NK cells alone) from the degranulation detected in the NK cells/K562 co-culture.

### 2.7. Statistical Analysis

Statistical analysis was performed using the GraphPad Prism software v9. Flow cytometry data were analyzed using the FlowJo software, v10. Results are shown as mean ± SEM, one-way or two-way ANOVA, followed by Tukey’s post hoc test. *p*-values (*p*) ≤ 0.05 were considered statistically significant.

## 3. Results

### 3.1. TGFβ Is a Crucial Regulator of the Induction of Decidual-like NK Cells Activity Compared to IL-6

TGFβ, a major cytokine present in the tumor micro (tissue-local) and macro (circulating) environments in cancer patients, has been reported to induce the generation of anergic and decidual-like NK cells. To corroborate the notion that TGFβ acts as a major tumor-associated cytokines in the generation of decidual-like/anergic NK cells, we further tested the effects of TGFβ on NK cell polarization, as compared to IL-6, another relevant cytokine present in the tumor micro- and macro-environment. We confirmed our previously published data [33] and observed no differences in CD56^bright^ NK cell increase when compared TGFβ with IL-6 (Appendix A), while NK cells exposed to TGFβ had increased expression of CD9, compared with not treated and IL-6-exposed NK cells (Appendix A). Additionally, TGFβ was more effective than IL-6 in increasing the frequency of CD16-NK cells and in downregulating NKG2D, this latter in a statistically significant manner (Appendix A). Finally, TGFβ significantly reduced NK cell degranulation capabilities against K562 cells, as compared to IL-6 (Appendix A).

### 3.2. TIMP-1 and TIMP-2 Counteract the Generation of TGFβ-Induced Decidual-like NK Cells

Recently, TIMP1 has also been identified as a new immune-regulators/immune-checkpoint modulator [9,11,30]. In the present study, we evaluate the impact of TIMP1 and TIMP2 during NK cell polarization to a pro-angiogenic phenotype by TGFβ. We investigated the ability of the administration of TIMP1 and TIMP2 to limit the polarization of cytolytic NK cells towards pro-angiogenic/decidual-like NK cells. We used healthy donor-derived NK cells, co-treated for 72 h with TGFβ (10 ng/mL), and found that TIMPs were effective in reducing the percentage of CD56^bright^CD16^−^ decidual-like NK cells (Figure 1A,B). The gating strategy is shown in Appendix A.

Furthermore, we investigated the ability of TIMP1 and TIMP2 to modulate the expression of the CD9 and CD49a decidual markers, induced by TGFβ exposure. We observed that CD9 (Figure 2A) and CD49a (Figure 2B) surface antigen expression was elevated by TGFβ treatment, as previously reported [46,47] for healthy control-derived NK cells and was decreased following 72 h of co-treatment with TGFβ plus TIMP1 or plus TIMP2. As a proof of concept, we investigated whether conditioned media from CC/CRC cell lines (HT-19 and CaCo2) can induce a polarizing effect, as compared to TGFβ. We found in preliminary experiments that CC-conditioned media were less efficient in generating decidual-like NK cells (Appendix A). In polarization experiments, TGFβ was confirmed as a better inducer for the dNK-like polarization than IL6 (Appendix A).

We observed that TIMP1 and TIMP2 have a trend of reducing the frequency of CD49a^+^NK cells when co-administered with CC CM (Appendix A).

### 3.3. TIMP-1 and TIMP-2 Modulate the Expression of Activation and Exhaustion Markers in TGFβ-Induced Decidual-like NK Cells

NK cells isolated from cancer patients have been found to acquire anergic and exhausted phenotypes. TGFβ has been reported to participate in this process. Here, we tested the ability of TIMP-1 or TIMP-2 to limit the generation NK cells endowed with anergic/exhausted phenotypes. Both TIMP1 and TIMP2 were able to counteract the effects of TGFβ-induced NK polarization by enhancing NKG2D activation receptor (Figure 3A) and decreasing the exhaustion marker TIM-3 (Figure 3B) and PD-1 (Appendix A). Internal controls for NK cell degranulation capability (Appendix A).

### 3.4. TIMP-1 and TIMP-2 Modulate the Degranulation Abilities in TGFβ-Polarized NK Cells

Since we observed the ability of TIMPs to restore NKG2D expression in TGFβ-induced decidual-like NK cells, together with attenuation of TIM-3 and PD-1 levels, we tested the effects of TIMPs on NK cell degranulation abilities on K562 cells. TIMP 1 and TIMP2 alone did not impair degranulation marker CD107a. We found that TIMP-1 had a positive significant capability in counteracting TGFβ. (Figure 4). TIMP-2, however, only showed a trend, albeit not significant, in reverting the degranulation capabilities, when combined with TGFβ (Figure 4).

## 4. Discussion

ILCs (innate lymphoid cells) represent a recently identified heterogeneous family of mononuclear hematopoietic cells found mostly in solid tissues [21,22]. Based on their lymphoid morphology, surface antigens, transcription factor expression and cytokine production (Th1, Th2 and Th17-like), ILCs have been classified into three major groups, termed ILC1, ILC2 and ILC3 [22]. Whether ILCs can be defined as friends or foes in cancer insurgence and progression is still a matter of debate [21,22,24]. NK cells are innate lymphoid cells involved in tumor immunosurveillance by inducing cancer cell lysis either directly (via perforin/granzyme system) or indirectly (via secretion of TNFα and IFNγ) [21,25]. NK cells develop altered immune-suppressive phenotypes in many cancer types.

Very recently, we found that in peripheral blood from prostate cancer (PCa) patients, NK cells show enhanced CD9, CD49a and CXCR4 expression [33]. PCa TANKs produce factors that are able to support inflammatory angiogenesis in an in vitro model and increase the expression of CXCL8, ICAM-1 and VCAM-1 mRNA in ECs. Secretome analysis revealed the ability of PCa TANKs to release pro-inflammatory cytokines/chemokines involved in monocyte recruitment and M2-like polarization [33].

Apart from anergy, we found that tumor-associated NK cells (TANKs) also in non-small cell lung cancer (NSCLC), pleural effusions and colorectal cancers (CRC) acquire pro-angiogenic phenotype and functions [29,30,31,32,33]. We observed that tumor-associated (TANKs) and tumour-infiltrating (TINKs) NK cells in cancer patients can acquire phenotype and functions similar to NK cells in the developing decidua [34,35,36,37]. The dNK cells are characterized by a CD56^superbright^CD16^neg^ phenotype and by the presence of CD9 (a member of the tetraspanin protein family that plays a role in cell adhesion and cell motility) and CD49a (an integrin alpha 1 subunit that binds collagen and laminin) [35,38]. These dNK cells are poorly cytotoxic and are necessary to remodel spiral arteries during pregnancy [34,35,39,40]. Within the decidua, NK cells have a critical role in the tolerance of the developing fetus, the modulation of angiogenesis and decidualization, all necessary processes associated with pregnancy. dNKs produce angiogenic factors: VEGF, PlGF and IL-8 (CXCL8) [35,39,40,41], angiogenin [40,41] and angiopoietin 1 and 2 [35,41]. They also support tissue remodeling through the expression of matrix metalloproteinases (MMP-9). MMPs and TIMPs expression were observed in both decidual tissues and in dNK cells [42]. It was found that a wide variety of MMPs are expressed at the human fetal–maternal interface, supporting the highly invasive cellular activity and ECM remodeling that occurs during placental development.

Considering the intriguing dual role of TIMP1 in regulating pathways of cancer progression in a bi-phasic manner as described previously [2,5,6,7] and recent evidence that TIMPs can function as alarmins [10,11,12], we examined the effects on ILCs as a focus for our current study. Recently, TIMP1 has also been identified as a new immune-regulator/immune-checkpoint modulator [9,11,30]. In the present study, we evaluate the impact of TIMP1 and TIMP2 during NK cell polarization to a pro-angiogenic phenotype by TGFβ. Subsequently, our goal was to determine how extracellular TIMPs can influence tumor-cell-polarized NK cell phenotype and behavior.

Interestingly, some NKG2D ligands (MICA, MICB, ULBP-2 and ULBP-3) are secreted from NSCLC [33], osteosarcomas [44] and gastric [45] cancer cells and may be shed by MMP-mediated cleavage. For example, upregulation of MMP activity can induce a downregulation of expression of NKG2D ligands in gastric cancer cells, leading to lower-level cancer susceptibility of NK cells [45]. Therefore, TIMPs could be relevant in NK cell-recognition activities.

Our current observations suggest that TIMPs can provide important new markers for the diagnosis, classification, prognosis and guide to treatment of the inflammation cells, and common and often aggressive forms of human cancer, such as CRC and PCa.

NK cells from colorectal cancer patients are polarized toward a pro-angiogenic phenotype, and they express angiogenin, MMP2 and MMP9 [30]. We also report that they paradoxically produce higher levels of TIMP-1 and TIMP-2 [30]. TGFβ mimics the cancer microenvironment. We found that NK cells from healthy donors, when exposed to TGFβ, acquire the CD56^bright^CD9^+^CD49a^+^ decidual-like phenotype, together with decreased levels of the activation marker NKG2D. Administration of TIMP1 or TIMP2 proteins was effective in interfering with TGFβ-induced NK cell polarization towards the CD56^bright^CD9^+^CD49a^+^ decidual-like phenotype and in the restoration of the levels of the NKG2D activation marker. We found that TIMP-1 was able to restore degranulation capabilities when combined with TGFβ, while TIMP-2 was not significantly efficient in limiting the TGFβ effect.

In preliminary experiments, we explored the polarizing activities of IL-6, another cytokine present in cancer patients, as compared to TGFβ. The IL-6/JAK/STAT3 signaling pathway is aberrantly hyperactivated in patients with hematopoietic malignancies or solid tumors in cancer patients [54,55]. In cytolytic NK cells, we observed that IL-6 was not effective in generating CD56^bight^CD9^+^CD49a^+^ NK cells and did not contribute to the generation of anergic/exhausted NK cells, as TGFβ does.

TGFβ has been shown by us and several authors to be a very important protumorigenic, pro-inflammatory and pro-angiogenic factor. We and others have shown how TGFβ is crucial in NK cells’ pro-angiogenic decidual-like polarization.

Our results suggest a potential role of TIMPs in controlling the pro-angiogenic switch induced by TGFβ in NK cells and suggest a possible use of TIMPs peptides in the re-education of anergic/pro-angiogenic NK cells.

## 5. Conclusions

Our results provided the rationale that TIMPs can be envisaged as a regulator of NK cells by interfering with TGFβ-induced NK cell polarization and by contrasting the generation of pro-tumor/decidual-like NK cells within the tumor microenvironment. Additionally, the degranulation marker CD 107 decreased by TGF-beta is restored by TIMP1. Therefore, we propose the use of exogenous TIMPs as a potential strategy in the re-education of pro-angiogenic, TGFβ polarized NK cells, in synergy with agents able to increase degranulation capabilities.

## Figures and Tables

**Figure 1 cancers-13-04955-f001:**
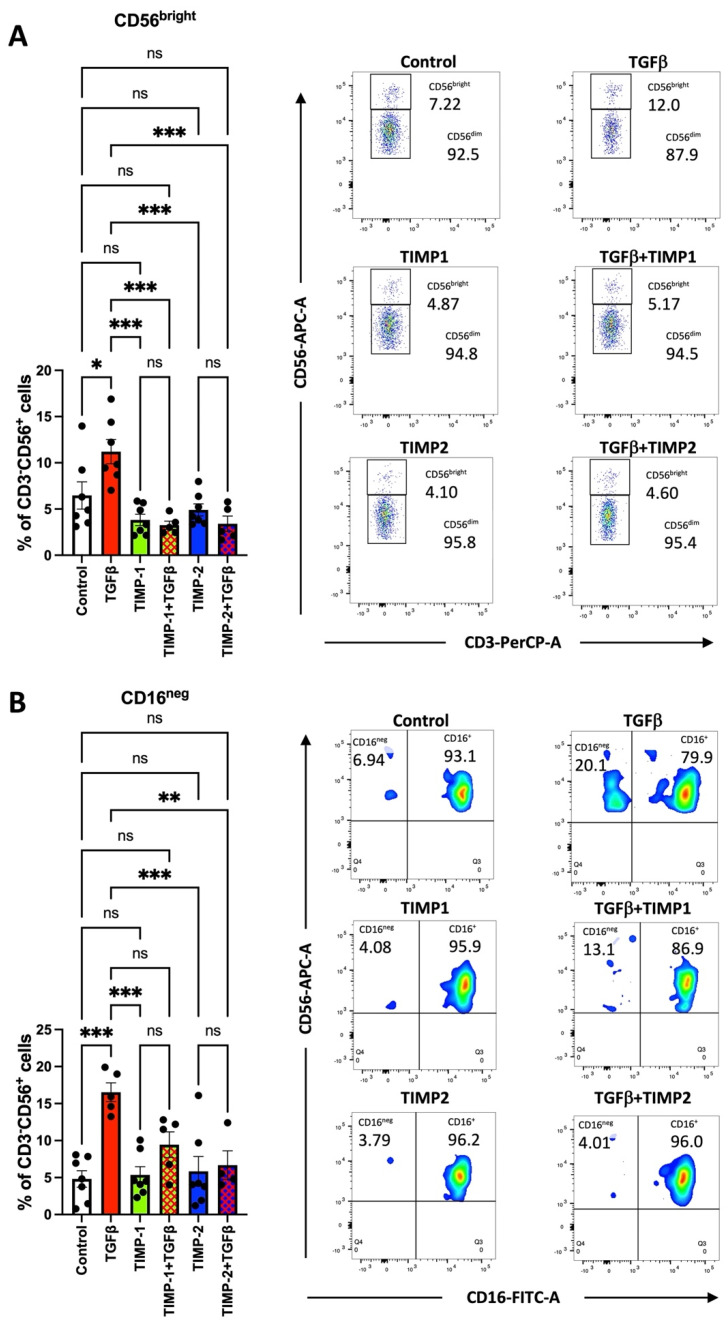
Healthy donor-derived NK cells exposed to TGFβ upregulate CD56^Bright^ (panel **A**) and CD16^neg^ (panel **B**), and TIMP1 and TIMP2 were effective in interfering with TGFβ. Results are shown as ± SEM, ANOVA, ns = not signficant; * *p* < 0.05; ** *p* < 0.01; *** *p* < 0.005. Representative plots are shown. Controls were cells in RPMI medium alone.

**Figure 2 cancers-13-04955-f002:**
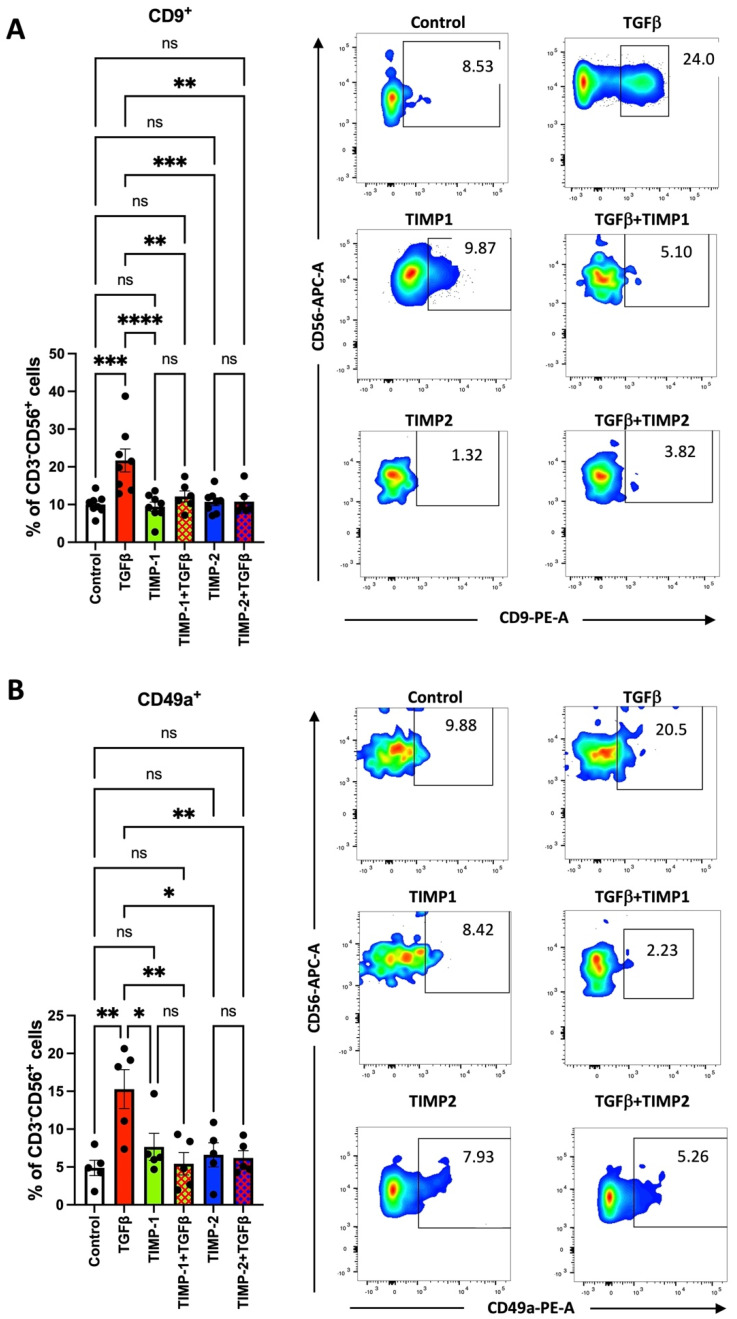
TGFβ in the polarization of healthy donor-derived NK TIMP1 and TIMP2 cells lowers the CD9 (panel **A**) and CD49a (panel **B**) levels (dNK markers) induced by TGFβ. Results are shown as mean ± SEM, ANOVA, ns = not signficant; * *p* < 0.05; ** *p* < 0.01; *** *p* < 0.005; **** *p* < 0.001. Representative plots are shown. Controls were cells in RPMI medium alone.

**Figure 3 cancers-13-04955-f003:**
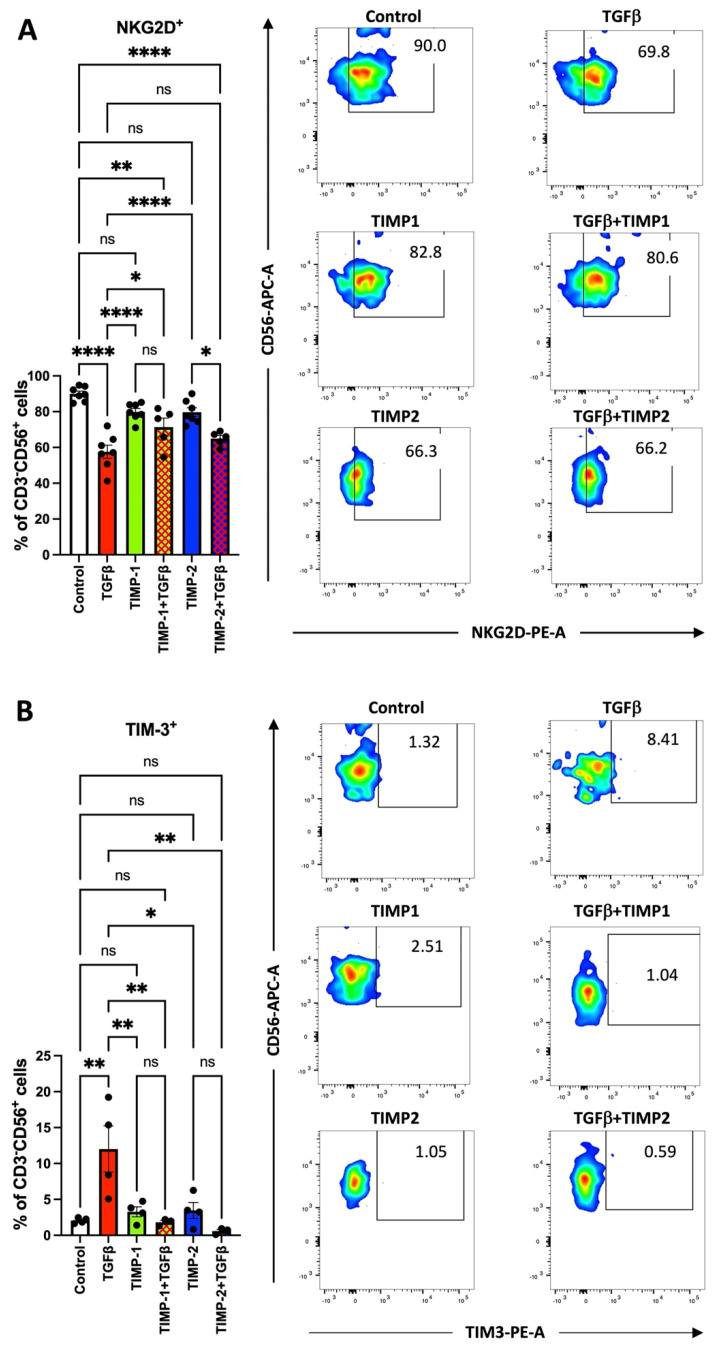
TIMP-1 and TIMP-2 treatments were effective in contrasting the TGFβ-induced ability to decrease the surface expression of the NK cell activation marker NKG2D (panel **A**) and the expression of the TIM-3 (panel **B**) exhaustion marker. Results are shown as mean ± SEM, ANOVA, ns = not signficant; * *p* < 0.05; ** *p* < 0.01; **** *p* < 0.001. Representative plots are shown. Controls were cells in RPMI medium alone.

**Figure 4 cancers-13-04955-f004:**
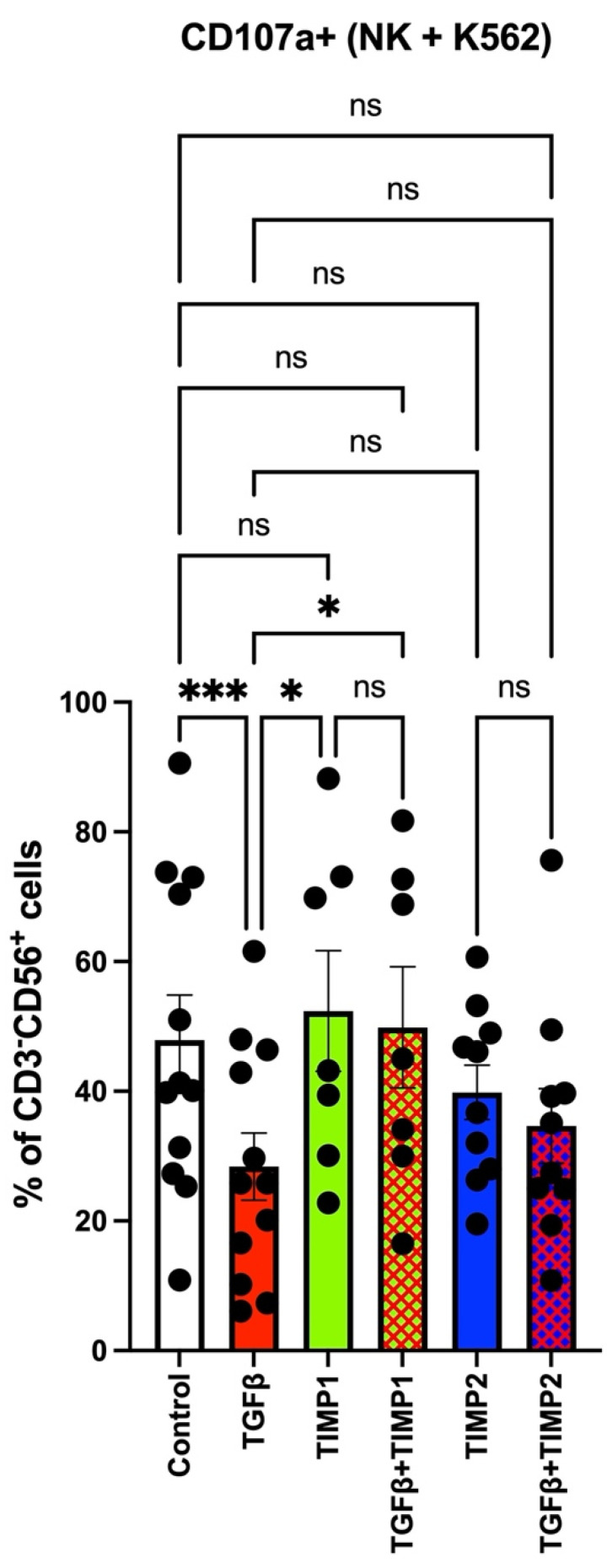
Effects of TIMPs on degranulation capabilities in TGFβ-induced decidual-like NK cells. TIMP-1 was significantly different from TGFβ, and TIMP-1 when combined TGFβ was able to counteract TGFβ. TIMP-2 did not have a significant ability to revert the impaired degranulation capabilities when combined with TGFβ. The graph shows the specific degranulation in the co-culture system with subtracted basal CD107a levels (in NK cells alone/in absence of cell target). Results are shown as mean ± SEM, ANOVA, ns = not signficant; * *p* < 0.05; *** *p* < 0.005. PBMCs from 7–12 different healthy donors were used. Controls were cells in RPMI medium alone.

## Data Availability

The data presented in this study are available on request from the corresponding author.

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
