# Peer review of "TIMP1 and TIMP2 Downregulate TGFβ Induced Decidual-like Phenotype in Natural Killer Cells"

_cancers, 2021, doi:10.3390/cancers13194955_

Round 1

Reviewer 1 Report

In “TIMP1 and TIMP2 downregulate TGFb induced decidual-like phenotype in Natural killer cells” by Albini et al., the authors investigated how tissue inhibitors of metalloproteases 1 and 2 (TIMP1-2) regulate the decidual-like phenotype of human Natural Killer (dNK) cells isolated from healthy donors. dNK are defined as CD56highCD16-CD9+CD49a+ that are poorly cytotoxic and are necessary for angiogenesis and decidualization during pregnancy. Indeed, dNKs produce both angiogenic factors and the matrix metalloproteinases MMP9 and MMP2, which are important for the breakdown of extracellular matrix required for vascular remodeling and trophoblast invasion. Several studies have indicated that some tumor infiltrating NKs acquires a dNK phenotype that supports tumor growth and progression. Several cytokines/stimuli, such as TGF-beta and hypoxia, can contribute to dNK polarization.

In the present manuscript and previous publications, the authors indicated that TIMP1-2 is able to counteract the dNK polarization induced by TGF-beta.

The study proposes a novel role of TIMP1-2 in dNK polarization, and might be of potential high impact in the field cancer immunology and cancer immunotherapy.

However, the lack of functional and cancer-related experiments overshadows the strength of the study and prevents it to be accepted for publication in the current form.

For this reason, I strongly recommend to include key additional experiments to increase the relevance of the study and to better adhere to the scope of Cancers.

Major points:

  1. In the “materials and methods”, the title of section 2.2. is “Isolation of mononuclear cells from whole blood of heathy donors and CRC patients”. However, there is no other mention nor use of the CRC samples in the study. Why not including NK from cancer patients? Perhaps, the authors might show whether dNKs from CRC patients and healthy donors differ at steady state and upon TGF-beta polarization +/- TIMPS. Can TIMP1-2 treatment rescue the dNK phenotype also in CRC samples?

  1. The authors show that both TIMP1 and 2 are able to revert the TGF-beta-induced dNK phenotype, analyzed as increased NKG2D and decreased TIM-3 levels. Even though these two markers are bona fide NK activation and exhaustion markers, they are not sufficient alone to prove that TIMP1-2 induce a functional switch of the NK cells. The authors should perform other functional assays +/- TIMP1-2, including NK proliferation, activation/degranulation (e.g. IFN-gamma, granzyme B, CD107a), ability to kill cancer cells (ADCC and not-ADCC) or inhibit cancer cell migration.

  1. Immune checkpoint inhibitors (ICIs) have been shown to induce NK anti-tumor response. Does PD-1/PD-L1 expression changes upon dNKs polarization and TIMP1-2 rescue? Does TIMP1-2 treatment enhance the ICI-induced NK response?

Minor points:

  1. The introduction is way too long and gives far too much information. On the contrary, the discussion is somewhat hasty and it does not add a deeper perspective of the study. Perhaps, some sections of the introduction should be moved to the discussion and integrated with the new experiments. Also, my advice is to focus more on the cancer-related aspects that is the main topic of Cancers

  1. The authors mentioned in the discussion that they have preliminary results regarding NK from individuals affected by BPH, an inflammatory condition predisposing to prostate cancer, in which TIMP1 expression seem to be high. The authors conclude that TIMP1 activation is acting as danger and control signal. However, one should argue that elevated levels of TIMP1 (but not TIMP2) predicts poor prognosis in many cancers, as also summarized by the authors in the introduction of the manuscript. In this study, TIMP1-2 seem to have the same effect on the phenotype of NK cells. How can the authors reconcile these apparently contrasting results? Please, comment on that.

  1. Number of biological samples and replicates should be added in each figure and/or legend. Bar graphs should contain dots indicating each sample/replicate and both error limits (above and below) should be indicated. “Results are shown as ± SEM” is not correct. Results are usually indicated as median or mean ± SEM. Please, add a statistical analysis paragraph in the “materials and methods” section. Even though ANOVA is indicated in the figure legends, the authors do not specify whether it was one-way or two-way ANOVA. Moreover, pair-wise comparisons are indicated with brackets in the graphs, but there is no mention about which type of multiple comparisons was performed.

  1. Table 1 (summarizing antibodies) is missing. Either insert the table or summarize clone/cat number/RRID of the relevant antibodies in the text.

  1. Y axis label of bar graphs is confusing: it should be CD3-CD56+ cells (% of total or % of gated). Also make X axis labels consistent in all figures: for example, sometimes it is TGF-beta 10 ng/ml and sometimes it is TGFb. The dot plots in Figure 3 lower panel need sample names.

  1. It would be nice to see an example of the gating strategy for all the experiments, maybe in supplementary. In Figure 2, lower panel dot plots and figure 3 upper panel dot plots: there seems to be populations stuck at the upper X axis. In both figures, the gate % take those populations into account. Can you please comment on that? If those are part of the positive gates, the samples should have been compensated differently. Otherwise, I would remove these population from the count and update the gates accordingly.

  1. Few typos.

Line 27 NK cells.

Line 28 add “and increased levels of TIM-3”

Line 51 a space is missing before ~40%

Line 132 “Our finding suggests”.

In general, all acronyms (e.g. dNK, TME, PCa, BPH etc) should be spelled out only the first time they are mentioned. Please, update them accordingly throughout the manuscript. 

Author Response

Comments on Reviewer 1

In “TIMP1 and TIMP2 downregulate TGFb induced decidual-like phenotype in Natural killer cells” by Albini et al., the authors investigated how tissue inhibitors of metalloproteases 1 and 2 (TIMP1-2) regulate the decidual-like phenotype of human Natural Killer (dNK) cells isolated from healthy donors. dNK are defined as CD56highCD16-CD9+CD49a+ that are poorly cytotoxic and are necessary for angiogenesis and decidualization during pregnancy. Indeed, dNKs produce both angiogenic factors and the matrix metalloproteinases MMP9 and MMP2, which are important for the breakdown of extracellular matrix required for vascular remodeling and trophoblast invasion. Several studies have indicated that some tumor infiltrating NKs acquires a dNK phenotype that supports tumor growth and progression. Several cytokines/stimuli, such as TGF-beta and hypoxia, can contribute to dNK polarization.

In the present manuscript and previous publications, the authors indicated that TIMP1-2 are able to counteract the dNK polarization induced by TGF-beta.

The study proposes a novel role of TIMP1-2 in dNK polarization and might be of potential high impact in the field cancer immunology and cancer immunotherapy.

However, the lack of functional and cancer-related experiments overshadows the strength of the study and prevents it to be accepted for publication in the current form.

For this reason, I strongly recommend including key additional experiments to increase the relevance of the study and to better adhere to the scope of Cancers

Response: We thank the reviewer for this comment. We have revised the paper according to the requests.

 Major points:

1. In the “materials and methods”, the title of section 2.2. is “Isolation of mononuclear cells from whole blood of heathy donors and CRC patients”. However, there is no other mention nor use of the CRC samples in the study. Why not including NK from cancer patients? Perhaps, the authors might show whether dNKs from CRC patients and healthy donors differ at steady state and upon TGF-beta polarization +/- TIMPS. Can TIMP1-2 treatment rescue the dNK phenotype also in CRC samples?

Response: We validated our results of TGFb polarized NK cells and the effects of TIMP1 or TIMP2 on polarization induced by two different CRC cell lines (HT-29, HCT116) In a future papers we will investigate patients with cancer patients (CRC) which will require several months.

2. The authors show that both TIMP1 and TIMP2 are able to revert the TGF-beta-induced dNK phenotype, analyzed as increased NKG2D and decreased TIM-3 levels. Even though these two markers are bona fide NK activation and exhaustion markers, they are not sufficient alone to prove that TIMP1-2 induce a functional switch of the NK cells. The authors should perform other functional assays +/- TIMP1-2, including NK proliferation, activation/degranulation (e.g. IFN-gamma, granzyme B, CD107a), ability to kill cancer cells (ADCC and not-ADCC) or inhibit cancer cell migration.

Response: Following the reviewer’s suggestion, we performed degranulation assay (CD107a) against K562, using decidual-like NK cells exposed to TGFb w/wo TIMP1 or TIMP2.

3. Immune checkpoint inhibitors (ICIs) have been shown to induce NK anti-tumor response. Does PD-1/PD-L1 expression changes upon dNKs polarization and TIMP1-2 rescue? Does TIMP1-2 treatment enhance the ICI-induced NK response?

Response: We extended our panel of antigens modulated by TIMP1 and TIMP2 on TGFb polarized NK cells, including the surface expression of PD-1. We found that TIMP1 and TIMP1 decrease the PD-1 levels on TGFb-polarized NK cells.

Minor points:

1. The introduction is way too long and gives far too much information. On the contrary, the discussion is somewhat hasty, and it does not add a deeper perspective of the study. Perhaps, some sections of the introduction should be moved to the discussion and integrated with the new experiments. Also, my advice is to focus more on the cancer-related aspects that is the main topic of Cancer

Response: We edited the introduction. Discussion has been integrated, also commenting the new results generated during the revision.

2. The authors mentioned in the discussion that they have preliminary results regarding NK from individuals affected by BPH, an inflammatory condition predisposing to prostate cancer, in which TIMP1 expression seem to be high. The authors conclude that TIMP1 activation is acting as danger and control signal. However, one should argue that elevated levels of TIMP1 (but not TIMP2) predicts poor prognosis in many cancers, as also summarized by the authors in the introduction of the manuscript. In this study, TIMP1-2 seem to have the same effect on the phenotype of NK cells. How can the authors reconcile these apparently contrasting results? Please, comment on that.

Response: This apparent paradox is commented in introduction and discussion.

3. Number of biological samples and replicates should be added in each figure and/or legend. Bar graphs should contain dots indicating each sample/replicate and both error limits (above and below) should be indicated. “Results are shown as ± SEM” is not correct. Results are usually indicated as median or mean ± SEM. Please, add a statistical analysis paragraph in the “materials and methods” section. Even though ANOVA is indicated in the figure legends, the authors do not specify whether it was one-way or two-way ANOVA. Moreover, pair-wise comparisons are indicated with brackets in the graphs, but there is no mention about which type of multiple comparisons was performed.

Response: The number of biological samples and replicated have been detailed in figure 1, 2 and 3 single dots of reference are reported in the bar graph. The statistical analysis section has been integrated and revised.

4. Table 1 (summarizing antibodies) is missing. Either insert the table or summarize clone/cat number/RRID of the relevant antibodies in the text.

Response: Clones for antibodies used for flow cytometry have been indicated in the dedicated method section

5. Y axis label of bar graphs is confusing: it should be CD3-CD56cells (% of total or % of gated). Also make X axis labels consistent in all figures: for example, sometimes it is TGF-beta 10 ng/ml and sometimes it is TGFb. The dot plots in Figure 3 lower panel need sample names.

Response: We fixed the figures

6. It would be nice to see an example of the gating strategy for all the experiments, maybe in supplementary. In Figure 2, lower panel dot plots and figure 3 upper panel dot plots: there seems to be populations stuck at the upper X axis. In both figures, the gate % take those populations into account. Can you please comment on that? If those are part of the positive gates, the samples should have been compensated differently. Otherwise, I would remove these population from the count and update the gates accordingly.

Response: We showed a representative gating strategy for NK cell selection. The gate have been placed in the positive cell populations, based on the compensation applied, also considering that cells remain in culture for 72 hours.

7. Few typos.

Line 27 NK cells.

Line 28 add “and increased levels of TIM-3”

Line 51 a space is missing before ~40%

Line 132 “Our finding suggests”.

Response: We corrected the typos

In general, all acronyms (e.g. dNK, TME, PCa, BPH etc) should be spelled out only the first time they are mentioned. Please, update them accordingly throughout the manuscript. 

Response: We have acronyms spelled out the first time.

Reviewer 2 Report

Authors found that NK cells in the peripheral blood of colorectal cancer patients overexpress TIMP1 and TIMP2. Healthy donors NK cells were exposed to TGFb, and characterized them using multicolor flow. Further, authors studied the effects of TIMP1 and TIMP2 on decidual like NK phenotypes generated in their study. Authors showed TIMP1 or TIMP2 administration was effective in interfering with TGFb induced NK cell polarization towards CD56brightCD9+CD49a+ decidual-like-phenotype and in restoration of the levels of the NKG2D and TIM-3. Based on their results, authors suggest that TIMPs regulate cancer associated NK cells.

Following comments to be addressed before considering this article for publication.

Comments:

Major points:

  1. Results needs to be validated using suitable alternative system/model.
  2. Results were described in short. Needs to be elaborated.
  3. Details were missing in figure legends Example: what is NT?, what does the bar graphs represent? What is X-axis in the bar graphs? etc..,

Minor points:

  1. Subdivide Figures into A and B.
  2. Page #3, line 62- 63, brain cancer was mentioned twice.
  3. Page #5, “2.3” is missing in the materials and methods.
  4. Font size is too small within the histograms of the multicolor flow cytometry results.  
  5. Maintain consistency, example: CD16- in page #6, line 192, while it was written as CD16neg in page #6, line 195.

Author Response

Reviewer 2

Authors found that NK cells in the peripheral blood of colorectal cancer patients overexpress TIMP1 and TIMP2. Healthy donors NK cells were exposed to TGFb and characterized them using multicolor flow. Further, authors studied the effects of TIMP1 and TIMP2 on decidual like NK phenotypes generated in their study. Authors showed TIMP1 or TIMP2 administration was effective in interfering with TGFb induced NK cell polarization towards CD56brightCD9+CD49a+ decidual-like-phenotype and in restoration of the levels of the NKG2D and TIM-3. Based on their results, authors suggest that TIMPs regulate cancer associated NK cells.

Following comments to be addressed before considering this article for publication.

Comments:

Major points:

1. Results needs to be validated using suitable alternative system/model.

Response: We thank the reviewer for this comment. Our revised paper includes different functional experiments (NK cell degranulation assay against K562 cells, following dNK cell polarization induced by two different CC cell lines (HT-29, HCT116) treated with TIMP1 or TIMP2.

2. Results were described in short. Needs to be elaborated.

Response: Results sections have been expanded.

3. Details were missing in figure legends Example: what is NT?, what does the bar graphs represent? What is X-axis in the bar graphs? etc..,

Response: Figure legends have been integrated

Minor points:

  1. Subdivide Figures into A and B.
  2. Page #3, line 62- 63, brain cancer was mentioned twice.
  3. Page #5, “2.3” is missing in the materials and methods.
  4. Font size is too small within the histograms of the multicolor flow cytometry results.  
  5. Maintain consistency, example: CD16in page #6, line 192, while it was written as CD16neg in page #6, line 195.

Response: All the minor points have been addressed.

Round 2

Reviewer 1 Report

Even though the authors tried to address the major concerns I raised during the first revision, there are still crucial conceptual and formal flaws which prevent the publication of the manuscript in its current form.

Major concern 1

CRC condition media (CM) production, CRC-CM polarization/rescue experiment and relative figure 4. Materials and methods section for CM production (2.2) says that CRC cells were starved in basal RPMI for 72h. I wonder why the authors decided to starve CRC cell lines, over a period of 72h. Starvation usually limits protein synthesis and cell homeostasis, therefore also cytokine production might be negatively affected as well. Second, CM is generally harvested within 24-48 hours to avoid excessive nutrients expenditure and acidification. This may be one of the reasons accounting for the not-significant results in the relative figure 4. A second issue is the polarization using this CM, which, first of all, fails to induce a significant decidual polarization in figure 4 (that is the required starting point to then investigate the effects of TIMPs). This may be due to the incorrect CM preparation (as explained above), incorrect CM/medium ratio used (the authors indicate 30% v/v, which may not be enough to induce polarization), a combination of both, or the inability of the CRC CM to induce dNK polarization. Therefore, I recommend to repeat the CM preparation in regular medium and harvested CM after shorter incubation times (24-48h). I also recommend to make a titration curve for the CM % necessary to induce a significant decidual-like polarization. This step is necessary to then move on with the TIMP-mediated rescue: without it, this experiment makes the article weaker rather than stronger, and disprove the previous results obtained with TGF-beta. With the current data authors cannot state the conclusions as they did, because results are not significant. Another issue might be that CM does not contain only TGF-beta: how can the authors ascertain that the effects they will see is mainly due to TGF-beta and not other secreted factors? If the new experiments will validate the data presented in figure 1-3, I would consider changing the title as a more general “TIMP1 and TIMP2 counteract the decidual-like polarization of Natural killer cells”.

Major concern 2

Degranulation assay and relative figure 5: the method section is not well detailed and referenced. There is a conceptual mistake in the effector:target (E:T) ratio used that may have negatively affected the significance of the experiment. As referenced elsewhere (e.g. Lorenzo-Herrero S., Sordo-Bahamonde C., Gonzalez S., López-Soto A. (2019) CD107a Degranulation Assay to Evaluate Immune Cell Antitumor Activity. In: López-Soto A., Folgueras A. (eds) Cancer Immunosurveillance. Methods in Molecular Biology, vol 1884. Humana Press, New York, NY. https://doi.org/10.1007/978-1-4939-8885-3_7), E:T is usually in a range of 10:1-5:1. The E:T ratio used here is 1:1, and that may explain the absence of significant results in figure 5. Also, the authors mentioned the use of positive and negative controls, but they make confusion about the required controls. First, they generally state “cells were stimulated for 6 hours with the cell stimulation cocktail (TONBO Biosciences)”. To my knowledge, this cocktail should be used only for the positive degranulation control (PBMC alone with PMA, Ionomycin, and the protein transport inhibitors Brefeldin A and Monensin). Basal degranulation control should contain only PBMC with complete medium: this means that there should be a basal degranulation control for each treatment in absence of K-562 cells. All the other samples, in which there is the E:T combo in different media, should not contain the stimulation cocktail: protein transport inhibitors (Brefeldin A and/or Monensin) are the only reagents to be added to the mix. I strongly recommend perform some optimization and repeat this experiment correctly. As it is, it does not improve the credibility of the study, instead it raises more doubts about its significance. With the current data authors cannot state the conclusions as they did, because the results are not significant.

For both revised new figure 4, figure 5 (experiments to be repeated) and supplementary figure 1 please show flow cytometry dot plots as in the previous figures. Please show dots of replicates on histograms. Include both error limits (above and below). Figure legends still need “mean” +/- SEM and type of ANOVA test. Consider adding “ns” for not significant comparisons.

Minor concerns

Authors said: “we showed a representative gating strategy for NK cell selection. The gate have been placed in the positive cell populations, based on the compensation applied, also considering that cells remain in culture for 72 hours”. However, I did not find this material in the revised version. Can the gating strategy be inserted in the supplementary, please? I would like to see an example of the entire population distribution (FSC and SSC) and then the gating from there.

Some typos

lines 125-140 “rationale”

line 134-135 fix sentence.

line 185 fix title

line 207 typos “at” and replace “following 72 hours” with “after 72 hours”

line 226 suggest replace “modulate” with either “counteract” or “limit”

lines 232-234 fix sentence

line 252 remove “diverse”

and others sentences need to be fixed along the text. Please, revise and correct accordingly.

Author Response

Even though the authors tried to address the major concerns I raised during the first revision, there are still crucial conceptual and formal flaws which prevent the publication of the manuscript in its current form.

Comments: We thank the reviewer for further comments. We did additional experiments, we have added a new figure, detailed the methods, implemented the References, and have modified the manuscript accordingly to the suggestions, as described point by point below.

Major concern 1

CRC condition media (CM) production, CRC-CM polarization/rescue experiment and relative figure 4. Materials and methods section for CM production (2.2) says that CRC cells were starved in basal RPMI for 72h. I wonder why the authors decided to starve CRC cell lines, over a period of 72h. Starvation usually limits protein synthesis and cell homeostasis, therefore also cytokine production might be negatively affected as well. Second, CM is generally harvested within 24-48 hours to avoid excessive nutrients expenditure and acidification. This may be one of the reasons accounting for the not-significant results in the relative figure 4.

Comments: We thank the reviewer for the observation. 72 hours was indeed a typo, there was a carry over of a sentence from a previous protocol. In this case indeed we used 48 hrs incubation with serum free medium (SFM), with no previous starvation time (AA did the incubation, apologies for the typo). This is being now corrected in the text. However, as for general consideration, in standard protocols for starvation, (meaning the use of medium without serum), 72 hours are proposed to observe a stronger biological effect. See for our published papers where the same protocol has been used (Gallazzi, et al. Front Immunol 2021, 11, 586126, doi:10.3389/fimmu.2020.586126).

Also, we previously published that the use of CM (from prostate cancer cell lines) 30% v/v is sufficient to obtain immune cell polarization (Gallazzi, et al. Front Immunol 2021, 11, 586126, doi:10.3389/fimmu.2020.586126.)

Serum free medium (starvation is a confounding word, we apologize) is used because ever after following de-complementation, FBS still contain growth factors that might add to those contained in the CM false the result. Starvation SFM will allow to observe biological effect only determined by the CM content.

A second issue is the polarization using this CM, which, first, fails to induce a significant decidual polarization in figure 4 (that is the required starting point to then investigate the effects of TIMPs). This may be due to the incorrect CM preparation (as explained above), incorrect CM/medium ratio used (the authors indicate 30% v/v, which may not be enough to induce polarization), a combination of both, or the inability of the CRC CM to induce dNK polarization. Therefore, I recommend repeating the CM preparation in regular medium and harvested CM after shorter incubation times (24-48h). I also recommend making a titration curve for the CM % necessary to induce a significant decidual-like polarization. This step is necessary to then move on with the TIMP-mediated rescue: without it, this experiment makes the article weaker rather than stronger, and disprove the previous results obtained with TGF-beta.

Comments: We apologize for not being clear enough in the method section describing the preparation of CM. CM preparation has been performed as follows. Cells were maintained in RPMI 1640 medium, supplemented with 10% Fetal Bovine Serum (FBS), (Euroclone), 2 mM l-glutamine (Euroclone), 100 U/ml penicillin and 100 μg/ml streptomycin (Euroclone). Once cells reached 80% of confluency, cells were washed 30 minutes in serum free RPMI medium, to eliminate serum residuals. Following cell layer wash, cells were maintained in serum-free RPMI medium for 48 hours and conditioned media were collected.

The aim of the paper was to show TGFbeta effects, so the use of the cancer cells CM was only to preliminarily propose that also cancer cell products can regulate TIMP expression, as preliminary data for further work. The CM data is now in supplementary material.

With the current data authors cannot state the conclusions as they did, because results are not significant. Another issue might be that CM does not contain only TGF-beta: how can the authors ascertain that the effects they will see is mainly due to TGF-beta and not other secreted factors? If the new experiments will validate the data presented in figure 1-3, I would consider changing the title as a more general “TIMP1 and TIMP2 counteract the decidual-like polarization of Natural killer cells”.

Comments: We thank the reviewer for this point. We published several papers demonstrating that TGFb acts as the master inducer dNK-like phenotype on cytolytic NK cells (Neoplasia 2013, J. of Imm. Res. 2018, F. in Immunology 2021). Here we wanted to see the effect of on the TIMPs a very crucial protein in regulating cancer on TGFb treated NK cells, to have a defined system.

In a recent publication, we demonstrated that other cytokines present in the tumor micro (tissue/local) and macro (peripheral blood environment), are not effective as TGFb in inducing the dNK-like phenotype. For example, we did not observe similar TGFb polarization effects when using IL-6 (F. in Immunology 2021, and current results). In our revision now we add a comparison to IL-6.

We confirmed that while TGFb acts as major inducer of dNK-like cells, IL6 does not. We think that adding this control inflammatory cytokine adds strengthen to detectable TGF beta shown for assessing TIMPs effects, and thank the reviewer for the comment.

Our observations strongly supports the idea of the parallel between the decidual/embryonic and tumor microenvironment that share the immunosuppressive/pro-angiogenic factor TGFb, as a hallmark.

Major concern 2

Degranulation assay and relative figure 5: the method section is not well detailed and referenced.

Comments: We used the same protocol in all our published papers, requiring the degranulation assay (J. of Imm. Res 2018, FASEB J. 2018, F. in Immunology 2021). We now made the method section more detailed and referenced.

There is a conceptual mistake in the effector:target (E:T) ratio used that may have negatively affected the significance of the experiment. As referenced elsewhere (e.g. Lorenzo-Herrero S., Sordo-Bahamonde C., Gonzalez S., López-Soto A. (2019) CD107a Degranulation Assay to Evaluate Immune Cell Antitumor Activity. In: López-Soto A., Folgueras A. (eds) Cancer Immunosurveillance. Methods in Molecular Biology, vol 1884. Humana Press, New York, NY. https://doi.org/10.1007/978-1-4939-8885-3_7), E:T is usually in a range of 10:1-5:1. The E:T ratio used here is 1:1, and that may explain the absence of significant results in figure 5.

Comments: As in the paper 10.1007/978-1-4939-8885-3_7 “Lysosome-associated    membrane    protein-1    (CD107a    or    LAMP-1) is a highly glycosylated transmembrane protein present in the lysosomes. In NK cells and cytotoxic T cells, CD107a is one of the most abundant proteins present in the lytic granules”, the degranulation assay is based on the detection of the CD107a antigen. The TONBO cocktail contains the necessary factors to avoid losing the CD107a by granule exocytosis, that is retained within the cell membrane and could be detected as surface antigen. We are now citing these relevant papers.

The use of the TONBO cocktail (or PMA, IONO, brefeldin A and/or monensin, as single agent, co-administered) is necessary to efficiently detect the CD107a as surface antigens (or eventually in combination with cytokine detection). We used the same protocol in our published paper, requiring the degranulation assay (J. of Imm. Res 2018, F. in Immunology 2021), with E:T 1:1. We choose the 1:1 ratio assuming that there is a single cell contact rather than 5-10 immune cells for every cancer cell.

Also, the authors mentioned the use of positive and negative controls, but they make confusion about the required controls. First, they generally state “cells were stimulated for 6 hours with the cell stimulation cocktail (TONBO Biosciences)”. To my knowledge, this cocktail should be used only for the positive degranulation control (PBMC alone with PMA, Ionomycin, and the protein transport inhibitors Brefeldin A and Monensin). Basal degranulation control should contain only PBMC with complete medium: this means that there should be a basal degranulation control for each treatment in absence of K-562 cells. All the other samples, in which there is the E:T combo in different media, should not contain the stimulation cocktail: protein transport inhibitors (Brefeldin A and/or Monensin) are the only reagents to be added to the mix. I strongly recommend perform some optimization and repeat this experiment correctly. As it is, it does not improve the credibility of the study, instead it raises more doubts about its significance. With the current data authors cannot state the conclusions as they did, because the results are not significant.

Comments: The degranulation assay is based on the detection of the CD107a antigen. The TONBO cocktail contains the necessary factors to avoid losing the CD107a by granule exocytosis, that is retained within the cell membrane and could be detected as surface antigen. Therefore, the use of the TONBO cocktail (or PMA, IONO, brefeldin A and/or monensin, as single agent, co-administered) is necessary to efficiently detect the CD107a as surface antigens (or eventually in combination with cytokine detection) in the control and treated cells. We explain this more clearly.

For both revised new figure 4, figure 5 (experiments to be repeated) and supplementary figure 1 please show flow cytometry dot plots as in the previous figures. Please show dots of replicates on histograms. Include both error limits (above and below). Figure legends still need “mean” +/- SEM and type of ANOVA test. Consider adding “ns” for not significant comparisons.

Comments: Former revised Figures 4 and 5 did not include representative dot plots, we consider the preliminary data to support future studies on colon cancer. Our paper is aimed to determine the role of TGB beta in TIMP 1 and 2 modulation. The figures are now in supplementary as supportive material.

Minor concerns

Authors said: “we showed a representative gating strategy for NK cell selection. The gates have been placed in the positive cell populations, based on the compensation applied, also considering that cells remain in culture for 72 hours”. However, I did not find this material in the revised version. Can the gating strategy be inserted in the supplementary, please? I would like to see an example of the entire population distribution (FSC and SSC) and then the gating from there.

Comments: We thank for the observation, we apologize for the missed representative figure showing the gating strategy, now included in revised Supplementary Figure 1.

Some typos

lines 125-140 “rationale”

line 134-135 fix sentence.

line 185 fix title

line 207 typos “at” and replace “following 72 hours” with “after 72 hours”

line 226 suggest replacing “modulate” with either “counteract” or “limit”

lines 232-234 fix sentence

line 252 remove “diverse”

and others sentences need to be fixed along the text. Please, revise and correct accordingly.

Comments: Typos has been checked and corrected.

Reviewer 2 Report

Recommended for publication.

Author Response

Comments and Suggestions for Authors

Recommended for publication.

We thank the reviewer.

Round 3

Reviewer 1 Report

Reviewer’s comment: The manuscript is improved, but still some major points are unresolved. Therefore I would reconsider the manuscript only if the authors are willing to fulfill at least the requested experiments/clarifications about CD107a. Also, after the 3rd round of review I expect that the authors will carefully amend all the fine minor errors/missing points requested since the 1st round.

Major concern 1

Reviewer’s comment: The fact the authors “previously published that the use of CM (from prostate cancer cell lines) 30% v/v is sufficient to obtain immune cell polarization (Gallazzi, et al. Front Immunol 2021, 11, 586126, doi:10.3389/fimmu.2020.586126.)“ does not exclude that a for different cancer cell line, 30% v/v might not be sufficient. It’s a pity that the authors ARE REFUSING to optimize these data, because they would be, perhaps, the most significant ones of the study, giving an additional value to the manuscript.

Major concern 2

Degranulation assay

Reviewer’s comments: Now the explanation is clearer. However, I still think that each population polarized differently, should have a basal degranulation control, to which the correspondent stimulated population should be normalized. In a reasonable in vivo setting, I imagine that even when the dNKs are polarized, the stimulus (e.g. TGFb and/or TIMPs) is still present. According to what I understand, TGF-beta and TIMPs, alone and in combination, are not present during the 6 hours of stimulation, right? If that is the case, I think this might also affect the degree of degranulation %. Can the authors comment on that? Did the authors assessed this point? I really appreciate the new figure 5. However, I think this should be your starting point and not the last figure, since your main focus should be the TIMPs effect at rescuing the phenotype induced by TGF-beta. Consider moving it as first figure or supplementary figure 1. Also, I cannot help it but notice a huge difference in the CD107a fold change between control and TGF-beta in figure 4 and figure 5. Indeed, while figure 5 has a significant phenotype with TGF-beta, figure 4 does not. Are the authors sure that the experiment in figure 4 worked? How can they explain the absence of significant inhibition of degranulation in the TGF-beta group? I would strongly recommend improve this point for the credibility of the study, and I still think that something did not work in this specific experiment. If the authors are not willing to repeat this experiment, they should tone down their conclusions, and they cannot say that one TIMP is more effective than another at mediating the degranulation capacity, since the results ARE NOT statistically significant mostly because TGF-beta polarization is not able to produce a significant phenotype in figure 4. They can only describe them as trends like they did for the CM experiments. In addition, for transparency, as I requested already multiple times, replicate dots and both limits of the error bars MUST be present in the graphs. Also, I would add in all the figure legends what exactly is the “control”, so that the reader does not get confused about it. Consider adding “ns” for all not significant comparisons.

Authors’ Comments: Former revised Figures 4 and 5 did not include representative dot plots, we consider the preliminary data to support future studies on colon cancer. Our paper is aimed to determine the role of TGB beta in TIMP 1 and 2 modulation. The figures are now in supplementary as supportive material.

Reviewer’s comments: Exactly, “Former revised Figures 4 and 5 (now in supplementary) did not include representative dot plots”. Therefore, the authors should add dot plots like they did for the other experiments. In addition, for transparency, as I requested already multiple times, replicate dots and both limits of the error bars MUST be present in the graphs. Also, I would add in all the figure legends what exactly is the “control”, so that the reader does not get confused about it.

Minor concerns

Reviewer’s comment: Some references are missing in the bibliography section. Please, update them correctly.

Author Response

Reviewer’s comment: The manuscript is improved, but still some major points are unresolved. Therefore I would reconsider the manuscript only if the authors are willing to fulfill at least the requested experiments/clarifications about CD107a. Also, after the 3rd round of review I expect that the authors will carefully amend all the fine minor errors/missing points requested since the 1st round.

We thank the reviewer for the comments on the improvement of our manuscript. We are resubmitting a third version of the manuscript addressing the concerns. New experiments as well as controls about CD107a assays are provided.

Novel experiments on the degranulation assay have been performed following the reviewer request. The related method section has been revised and detailed accordingly. For further clarity, we also added a Supplementary Figure to show the values for the internal controls for degranulation assay, in line with the reviewer’s indications.

Major concern 1

Reviewer’s comment: The fact the authors “previously published that the use of CM (from prostate cancer cell lines) 30% v/v is sufficient to obtain immune cell polarization (Gallazzi, et al. Front Immunol 2021, 11, 586126, doi:10.3389/fimmu.2020.586126.)“ does not exclude that a for different cancer cell line, 30% v/v might not be sufficient. It’s a pity that the authors ARE REFUSING to optimize these data, because they would be, perhaps, the most significant ones of the study, giving an additional value to the manuscript.

Our manuscript focuses on the effects of TIMPs on TGFbeta induced polarization, it is the first time that this property of TIMPs is studied and we envisaged a clean, simple biochemical question and model. However, in response to the reviewer, we performed new experiments by polarizing NK cell with CM of colon cancer cell lines HT-29 and colorectal cancer cell line CaCo2, with a determined quantity of total proteins in the CM.

We agree that using 30% v/v is rather vague. Working with supernatants (CMs) always introduce the problem or the real quantity of overall secreted product contained and the subsequent read out. In our revised manuscript, we used a determined and constant amount of CMs (here referred as total CM protein content/per polarization), to abate the bias of a non-controlled amount of total secrete product administered to NK cells. To minimize this uncertainty, we used 50 ug of total CM to pulse NK cells, thus administering a standardized amount of total protein.

Major concern 2

Degranulation assay

Reviewer’s comments: Now the explanation is clearer. However, I still think that each population polarized differently, should have a basal degranulation control, to which the correspondent stimulated population should be normalized. In a reasonable in vivo setting, I imagine that even when the dNKs are polarized, the stimulus (e.g. TGFb and/or TIMPs) is still present. According to what I understand, TGF-beta and TIMPs, alone and in combination, are not present during the 6 hours of stimulation, right? If that is the case, I think this might also affect the degree of degranulation %. Can the authors comment on that? Did the authors assessed this point?

In our in vitro model, the determination of impaired/restored degranulation represents the final read out of the induced polarization process. Therefore, during the degranulation, no TGFb and/or TIMPs are added, since we wanted to assess the final effects, following the polarization process.

According to the reviewer’s indications, we did not use the TOMBO kit  as a stimulus for our experimental conditions. Following NK cell polarization, the degranulation assay was performed following the reliever’s indications.

Following 72 hours of polarization, 2 × 105 MNCs were co-cultures with 2 × 105 K562 (E:T ratio of 1:1), in presence of anti-CD107a- FITC (BD Bioscience, H4A3), Golgi Plug (Bre-feldin, BD Biosciences) and Golgi Stop (Monesin, BD Biosciences). MNC alone were used as control to detect basal degranulation activities by NK cells, MNC treated with Iono-mycin (500 ng/mL, Sigma Aldrich), PMA (10 ng/mL, Sigma Aldrich), Golgi Plug (Bre-feldin, BD Biosciences) and Golgi Stop (Monesin, BD Biosciences), as positive control for non-specific degranulation, while K562 cell alone was use as internal control. This is reported in Material and Methods and there is a supplemental figure.

I really appreciate the new figure 5. However, I think this should be your starting point and not the last figure, since your main focus should be the TIMPs effect at rescuing the phenotype induced by TGF-beta. Consider moving it as first figure or supplementary figure 1.

We appreciate the suggestion and moved the result paragraph at the beginning and figure 5 to supplementary figure 1.

 Also, I cannot help it but notice a huge difference in the CD107a fold change between control and TGF-beta in figure 4 and figure 5. Indeed, while figure 5 has a significant phenotype with TGF-beta, figure 4 does not. Are the authors sure that the experiment in figure 4 worked? How can they explain the absence of significant inhibition of degranulation in the TGF-beta group? I would strongly recommend improve this point for the credibility of the study, and I still think that something did not work in this specific experiment. If the authors are not willing to repeat this experiment, they should tone down their conclusions, and they cannot say that one TIMP is more effective than another at mediating the degranulation capacity, since the results ARE NOT statistically significant mostly because TGF-beta polarization is not able to produce a significant phenotype in figure 4. They can only describe them as trends like they did for the CM experiments.

The reviewer is very right about this point. We repeated the experiment with the protocols suggested. In accordance with the reviver’ comments, we discussed as “trends” the modulation that are not related to statistically significance. In the new experiments showed in the revised manuscript, TGFb significantly worked in decreasing NK cell degranulation capability, when compared to NK cells treated with medium alone. TIMP1 and TIMP2 both did not decrease CD107a+ level. TIMP1 rescued inhibition by TIMP1, however TIMP2 was not significatively effective in counteracting TGFb action. Therefore, our conclusions are that TIMP1 and 2 can inhibit phenotype polarization of what are considered the decidual-like markers, and TIMP1 does also impact significantly on preserving degranulation capabilities,

 In addition, for transparency, as I requested already multiple times, replicate dots and both limits of the error bars MUST be present in the graphs. Also, I would add in all the figure legends what exactly is the “control”, so that the reader does not get confused about it. Consider adding “ns” for all not significant comparisons.

Replicate dots and both limits of the error bars are now present in the graphs.

The graph present in our revised version now shows results as bar with dots, to trace the number of healthy donors used per each experiment.

There is variability among primary NK cells from healthy donors, we made the effort to work with “real world” cells, instead of established cell lines, each donor has been used as single and each bar shows the average, per experimental group, of the donors used. We added “ns” for all non-statistically significant comparisons. We explained what the “controls” are cells receiving only  RMPI medium without TGFbeta or TIMPs.

Authors’ Comments: Former revised Figures 4 and 5 did not include representative dot plots, we consider the preliminary data to support future studies on colon cancer. Our paper is aimed to determine the role of TGB beta in TIMP 1 and 2 modulation. The figures are now in supplementary as supportive material.

Reviewer’s comments: Exactly, “Former revised Figures 4 and 5 (now in supplementary) did not include representative dot plots”. Therefore, the authors should add dot plots like they did for the other experiments. In addition, for transparency, as I requested already multiple times, replicate dots and both limits of the error bars MUST be present in the graphs. Also, I would add in all the figure legends what exactly is the “control”, so that the reader does not get confused about it.

See previous answer. We added dot plots and both limits of error bar, and clarified the figure legends for controls

Minor concerns

Reviewer’s comment: Some references are missing in the bibliography section. Please, update them correctly.

References have been checked and updated.